

# Research on three-state reliability evaluation method of high reliability system based on multi-source prior information

Jingde Huang[1], Zhangyu Huang[2] and Xin Zhan[3]

[1] Guangdong Intelligent Vision Precision Detection Engineering Technology Research Center, Zhuhai College of Science and Technology, Zhuhai, China
[2] Faculty of Innovation Engineering, Macau University of Science and Technology, Macau, China
[3] School of Mechanical Engineering, Zhuhai College of Science and Technology, Zhuhai, China

## ABSTRACT

A high reliability system has the characteristics of complexity, modularization, high cost and small sample size. Throughout the entire lifecycle of system development, storage and use, the high reliability requirements and the risk analysis form a direct contradiction with the testing expenses. In order to ensure the system, module or component maintains good reliability status and effectively reduces the cost of sampling tests, it is necessary to make full use of multi-source prior information to evaluate its reliability. Therefore, in order to evaluate the reliability of highly reliable equipment under the condition of a small sample size correctly, the equipment reliability evaluation model should be built based on multi-source prior information and form scientific computing methods to meet the needs of condition evaluation and fund assurance of high reliability system. In engineering practice, high reliability system or module gradually develops from normal state to failure state, generally going through three working states of "safety-potential failure-functional failure". Firstly, the historical test data under the three states can be used for the data source for the reliability evaluation of the system at the current stage, which supplements the deficiency of the field data; secondly, due to the lack of accurate judgment on the working state of a high reliability system or modules and analysis of the health status, the unnecessary maintenance may aggravate the evolution speed from potential failure to functional failure; thirdly, when high reliability system or module operates under overload or harsh conditions, the potential failure will be worsened to a certain extent. Aiming at the difficulty of multi-state system reliability evaluation, a reliability evaluation method based on non-information prior distribution is proposed by fusing multi-source prior information, which provides ideas and methods for reliability evaluation and optimization analysis of high reliability system or module. The results show that the three-state reliability evaluation method proposed in this article is consistent with the actual engineering situation, providing a scientific theoretical basis for preventive maintenance of high reliability system. At the same time, the research method not only helps evaluate the reliability state of a high reliability system accurately, but also achieves the goal of effectively reducing test costs with good economic benefits and engineering application value.

Corresponding author
Jingde Huang, jdh925@zcst.edu.cn, jdh925@sina.com

# INTRODUCTION

In the management activities of the whole life cycle of high reliability system, regular reliability sampling inspection is required at all stages. Considering that the limitation of small sample size and test cost are always the key factors which restrict the reliability evaluation. In order to make up for the shortage of test sample size, the historical test data or test identification data of similar products can also be used as the data source for evaluating the reliability of new prototype, providing a large amount of prior information for the sampling test, which ensures the best evaluation effect can be obtained at a small test cost. Besides, it maintains the equipment integrity rate and task success of the system in production, storage, use and other stages (*Zhang et al., 2005*). For high reliability system, in order to meet the task requirements of high load and other harsh environments, the fault detection method has become a necessary means to maintain the operation efficiency and good safety of high reliability system. Traditional maintenance methods (post maintenance, scheduled maintenance, preventive maintenance) cannot meet the requirements of power system when carry out tasks such as high load. Condition based maintenance has become an inevitable development trend. Effective fault diagnosis and prediction are prerequisite for condition based maintenance (CBM) (*Ahmed & Kabir, 2020*). In engineering practice, most faults are three-states process that gradually evolves from normal to failure, and these states change cannot be directly observed. Fault diagnosis, fault prediction and identifying the operating state of the high reliability system according to measured signals, can estimate state in the future, and predict the remaining service life of key modules. At present, the fault of high reliability system is mainly detected online by temperature, current, deformation and other sensors. The type of fault is determined by the reflection process of the fault point to the signal. However, due to the limitation of the measuring point or the unreachability of the measuring point and other factors, it is still very difficult to predict the early fault of power system. *Lavau, Suhrke & Knott (2023)* used a variety of sensors to complete the constraint control of state quantity, and realized the determination of control threshold; *Kim & Kim (2022)* used the improved mixed Gaussian clustering model and accelerated testing experiments to detect early faults in high reliability system, and predicted the faults that may be detected or occurred in the future; *Song et al. (2023)* detected and estimated the early fault condition of the motor by monitoring the current signal change of the inverter; *Wang et al. (2017)* established a new power system fault prediction model based on GM forecasting method. However, there are still many problems should be solved for potential failure prediction without failure data or obvious failure feature information during the use stage of equipment. Therefore, it is particularly urgent to explore the three-state reliability state evaluation method and potential failure detection technology of high reliability system.

Reliability centered maintenance analysis can confirm that the potential failure state has the characteristics of weak fault signal, unobvious fault signal characterization, little fault feature information, and many uncertain factors. Therefore, we can evaluate the potential failure state probability with the help of three-state reliability theory. However, the traditional fault diagnosis mainly aims at the situation where the fault mechanism is clear or there are many historical failure data (*Zhang et al., 2022*; *Babb & Rogatko, 2004*). For the potential failure prediction, it is difficult to locate the fault and determine the fault latency. In engineering practice, the symptom information of the initiation and expansion of potential failure in high reliability system has been reflected in its internal unit structure (*Feng et al., 2022*; *Mosleh et al., 2022*). Therefore, putting forward a scientific method for detecting potential failure in high reliability system, separating and extracting the characteristics of potential failure, and exploring the harm degree, the development law and the damage mechanism of potential failure will certainly contribute to the reliability state assessment of the high reliability system. Considering that Markov process states the transfer relationship between "current state" and "future state" of dynamic system from the perspective of probability. At the same time, Markov process effectively realizes the tracking of potential failure state, but in the process of state transition, there is a lot of uncertain factors, which are difficult to quantify accurately. Taking aerospace equipment as an example, aerospace equipment is a typical system with complexity, accuracy and high reliability. When the aerospace equipment is in a potential failure state, its external performance is weak, non-stationary and no substantial damage to the system. The detected data are "no failure" data, so the characteristic information of potential failure is relatively small, which belongs to typical small sample information. In order to accurately diagnose the latent failure of aerospace equipment with progressive development characteristics and no failure information, the quality status of aerospace equipment must be correctly evaluated. Therefore, the study of a set of reliability evaluation methods of aerospace equipment considers prior information is the premise of realizing the fault diagnosis of aerospace equipment. Besides, for aerospace equipment, simply relying on the whole machine operation and experience accumulation to obtain the diagnostic information of key modules cannot meet the requirements of equipment safety and reliability in terms of time and cost (*Ming et al., 2023*; *Huang, Duan & Hao, 2010*). Therefore, the correct evaluation of the reliability status of aerospace equipment will be helpful to the qualitative and quantitative analysis of potential failure laws. Take multi-electric/all electric aircraft as an example, the airborne power distribution system is very complex, and the number of aviation cables has increased significantly. Due to the limited internal space of the aircraft, thousands of cables are distributed and stacked in a cross manner, and have been working in high temperature; cold, strong ultraviolet radiation; vibration and other environments for a long time, resulting in gradual aging and damage of cables, arc failure, *etc*. However, the current airborne BIT is mostly aimed at explicit fault alarm, and the accuracy of early fault prediction is far from enough, which seriously affects social and personal safety. Therefore, it is urgent to study accurate fault prediction technology. Implicit or intermittent failure is an objective existence in aerospace equipment and highly integrated equipment, which has

obvious latency, intermittent and transient characteristics. It leads to the failure of timely protection of airborne TCBs and ECBs.

The traditional two-state (normal or failure) reliability theory can't explain the reliability problem of three-state complex system with progressive development characteristics (*Zhang, 2018*), especially for the acceptance test and sampling test of high reliability system. Those tests are typically small sample tests, the contradiction between small test samples and high reliability requirements directly affects the economic and social benefits (*Liu et al., 2020*). As we all know, the use of subjective experience has always been a controversial issue in reliability evaluation. However, in many cases, the use of subjective experience is an integral choice. The Bayes method does not exclude subjective experience, and regards it as a part of prior information. In engineering practice, the practical experience of experts and technicians is very valuable (*Ghosh, 2020*; *Cotterill, 2005*). Proper use of these experiences not only solve some theoretical difficulties, but also save a lot of test costs. Therefore, subjective experience shouldn't be excluded in engineering application, how to reasonably integrate these experiences and properly use them in specific problems should be studied. Due to the fact that multi-state reliability analysis more accurately reflects the actual working status of equipment, it has a wildly application in equipment fault diagnosis and prediction. In recent years, the author has conducted in-depth research on the role of multi-state reliability analysis in the state evaluation of high reliability system in the National Natural Science Foundation project. At the same time, the high value and the limited quantity of high reliability systems are difficult to collect sufficient information to accurately evaluate the reliability status of the system. Therefore, it is of great engineering significance to fully integrate multi-source information to make up for the shortcomings of on-site testing data and study the method of multi-source information fusion on reliability evaluation. This article studies the fusion method of prior information in three-state system, establishes a reliability evaluation model based on multi-source information fusion, and solves two problems related to binomial distribution: (1) how to determine the non-information prior distribution; (2) for multi-layer Bayes method, how to ascertain the super prior distribution by integrating expert experience. The solution of the above problems will help to reasonably evaluate the reliability status of the potential failure of the system and further identify the potential failure mode. The above methods and models are applicable to the system reliability assessment at all stages in the whole life cycle, and have positive engineering significance for saving test costs and ensuring the quality status of high reliability system.

## PROPOSED METHODOLOGY

### Bayes hypothesis of non-information prior distribution

The Bayes method puts forward that the unknown parameter is assumed to have equal probability within its possible range of values, that is when the unknown parameter is a continuous random variable, it is assumed to obey the uniform distribution in a certain interval (*Marichal, Mathonet & Paroissin, 2017*; *Ashrafi & Asadi, 2014*). If the unknown parameter is a discrete random variable with only a limited numerical value, it is assumed that the probability of taking these values is equal.

Criterion 1: for the success rate $R$ in binomial distribution:

According to the Bayes hypothesis and maximum entropy method, the prior distribution density is $Beta(1,1)$. On the basis of Jeffreys' criterion, its prior distribution density is $Beta(1/2,1/2)$. According to the method of Reformulation and ALI, the prior distribution density is $Beta(0,0)$.

In the above three kinds of non-information prior distributions, $Beta(0,0) = R^{-1}(1-R)^{-1}$ is not a normal density function, but a generalized prior distribution, and named prior distribution 1. $Beta(1/2,1/2)$ and $Beta(1,1)$ are normal density functions, named prior distribution 2 and prior distribution 3 respectively.

## Determination method of non-information prior distribution

The non-failure data source is a synthesis of machine data, information data and cross-border data of the industrial chain from all aspects of the equipment life cycle. In principle, the attributes of non-failure data sources belong to uncertain data, and their significant characterization feature is non-stationary. It is well known that the sharp changes of non-stationary signals are the most critical points in analyzing signals (*Le Guen & Thome, 2023*). For example, during the operation of precision electronic module, due to the erosion of insulation layer, acid, alkali, moisture and the role of cyclic stress, *etc.* It is very easy to catalyze the initiation and evolution of potential failure, thus leading to open circuit, short circuit, electric leakage, arc fault and other functional failures.

For the whole population (binomial distribution) of success or failure type test, the prior distribution of its distribution parameter $R$ (success rate) usually uses its conjugate prior distribution $Beta(a,b)$, so the problem becomes: how to determine the super parameters $a$ and $b$ without prior information. The usual practice is to take the prior distribution as $Beta(0,0)$, $Beta(1/2,1/2)$ or $Beta(1,1)$ (*Chien et al., 2023*; *Xu, Liu & Xiao, 2023*; *Stefan et al., 2022*). When solving specific engineering problems, how to determine which of them should be taken as a prior distribution is an important scientific problem.

### Comparison of several non-information prior distributions

In Criterion 1, we can see that the three non-information prior distributions have their own rationality in theory. It has been pointed out that no matter which one is used as a prior distribution, it has little impact on the results of Bayesian statistical indifference (*Coolen, Coolen-Maturi & Al-refaiee, 2014*; *Mkrtchyan, Podofillini & Dang, 2016*; *Jusn, Kim & Kim, 2013*). However, this is not necessarily the case.

Criterion 2: when the prior distribution is $Beta(0,0), Beta(1/2,1/2), Beta(1,1)$, respectively, for the given confidence $1-\alpha$ and test result $(s,f)$, the corresponding lower limit of reliability is $R_L^1$, $R_L^2$ and $R_L^3$ respectively, when $s > f$, there is a following formula:

$$R_L^1 > R_L^2 > R_L^3 \tag{1}$$

Here, if $s = 0$ or $f = 0$, then $R_L^1$ does not exist.

According to (*Huang, Duan & Hao, 2010*; *Wang, Cai & Jiao, 2007*), the differences between $R_L^1$, $R_L^2$ and $R_L^3$ are investigated through the simulation calculation. In each case of $R = 0.99, 0.95, 0.90, 0.85, 0.80, 0.75, 0.70, 0.65, 0.60, 0.55$ and $0.50$, we achieve a set of

random numbers conforming to binomial distribution according to the sample size $n = 5$, 8, 10, 15, 20, 30, 40, 60, 80, 100. For each group of 5,000 pieces, calculate the lower limit of reliability $R_L^1$, $R_L^2$ and $R_L^3$ according to three different prior distributions. Then calculate the average value and the risk ratio (percentage of $R_L^i > R$) of these 5000 $R_L^i$ pieces. The lower average value indicates that $R_L^i$ achieved is conservative. A higher risk ratio indicates that $R_L^i$ achieved risk tendency. Since $Beta(0, 0)$ is not applicable to zero failure, in order to facilitate comparative analysis, the generated random numbers are divided into two cases: non-zero failure and zero failure (all). The following table shows only some results. All calculations of the above methods are completed by using MATLAB, and the results are shown in Tables 1, 2 and 3.

As shown in Tables 1, 2 and 3:

(1) The difference of the reliability lower limit achieved by using different prior distributions that decreases with the larger sample size. After $n > 40$, the difference is not obvious; the sample size is smaller, the difference is more significant.

(2) When the reliability $R$ is high (more than 0.85) and the sample size is small (less than 30), it is more appropriate to use $Beta(0, 0)$ for reliability evaluation with prior distribution; When the reliability $R$ is low (less than 0.70) and the sample size is large (more than 40), the prior distribution should be $Beta(1, 1)$; When the $R$ is between 0.7 and 0.85, sample size is between 30 and 40, $Beta(1/2, 1/2)$ shall be adopted.

The above results can explain the reason for using different non-information prior distribution in engineering problems. For high reliability system, its layout, materials and technology are kept improving, and the preliminary tests are sufficient. In such conditions, the products should have high reliability while other systems may not be able to do so.

### *Application of fuzzy comprehensive evaluation*

When non-information prior distribution is adopted, the result reflects a prior understanding of the unknown entirety. The "non-information" doesn't mean ignorance. Through understanding the product design, materials, process, manufacturing and other aspects, we can always have a certain understanding of the reliability level of the product, but this understanding is empirical and subjective (*Goode & van de Lindt, 2013*; *Williamson, 2001*; *Lecoutre & Poitevineau, 2022*). If the reliability of the product is considered high enough, $Beta(0, 0)$ can be used; if the reliability of the product is considered medium level, $Beta(1/2, 1/2)$ can be used; when the reliability of the product is completely uncertain, $Beta(1, 1)$ should be adopted from a conservative perspective.

It can be difficult to divide people's prior understanding of product reliability into the above three levels. In actual use, it is easy to focus on the middle level, that is $Beta(1/2, 1/2)$. Therefore, two mixed beta distributions, $[Beta(0, 0) + Beta(1/2, 1/2)]/2$ and $[Beta(1/2, 1/2) + Beta(1, 1)]/2$ are constructed as uninformative prior distributions. The lower reliability limit achieved from them which is recorded as $R_L^4$ and $R_L^5$.

Criterion 3: for the given confidence level $1 - \alpha$ and test result $(s, f)$, $s$ represents the number of successful tests, and $f$ represents the number of failed tests. When $s > f$, there is a following formula:

$$R_L^1 > R_L^4 > R_L^2 > R_L^5 > R_L^3. \tag{2}$$

**Table 1  Differences in the lower limit of reliability obtained by three prior distributions for different sample sizes.**

| Average value all non-zero failure | Aggressive ratio all non-zero failure | $n = 5$ | | $n = 10$ | | $n = 20$ | | $n = 40$ | |
|---|---|---|---|---|---|---|---|---|---|
| prior distribution | 1 | 0.51184 | 0 | 0.70345 | 0 | 0.79571 | 0 | 0.83663 | 0.2129 |
| | | 0.65361 | 0 | 0.73805 | 0 | 0.79204 | 0.1218 | 0.82708 | 0.0792 |
| prior distribution | 2 | 0.47634 | 0 | 0.66517 | 0 | 0.77216 | 0 | 0.82474 | 0.0638 |
| | | 0.58077 | 0 | 0.69700 | 0 | 0.76971 | 0 | 0.51593 | 0.0792 |
| prior distribution | 3 | 0.45668 | 0 | 0.63688 | 0 | 0.75217 | 0 | 0.81377 | 0.0638 |

Notes.
When the reliability and confidence levels are taken as 0.9, the difference in the lower limit of reliability obtained by different sample sizes according to three prior distributions.

**Table 2  Differences in the lower limit of reliability obtained by three prior distributions for different sample sizes.**

| Average value all non-zero failure | Aggressive ratio all non-zero failure | $n = 5$ | | $n = 10$ | | $n = 20$ | | $n = 40$ | |
|---|---|---|---|---|---|---|---|---|---|
| prior distribution | 1 | 0.45038 | 0 | 0.60977 | 0 | 0.68384 | 0.1940 | 0.71789 | 0.1560 |
| | | 0.53817 | 0 | 0.61697 | 0.112 | 0.67101 | 0.0640 | 0.71046 | 0.0772 |
| prior distribution | 2 | 0.42712 | 0 | 0.58412 | 0 | 0.66806 | 0.0536 | 0.71041 | 0.0770 |
| | | 0.50068 | 0 | 0.59269 | 0.112 | 0.65770 | 0.0640 | 0.70343 | 0.0772 |
| prior distribution | 3 | 0.41538 | 0 | 0.56503 | 0 | 0.65505 | 0.0536 | 0.70338 | 0.0770 |

Notes.
When the reliability and confidence levels are taken as 0.8 and 0.9 respectively, the difference in the lower limit of reliability obtained by different sample sizes according to three prior distributions.

**Table 3  Differences in the lower limit of reliability obtained by three prior distributions for different sample sizes.** When the reliability and confidence levels are taken as 0.7 and 0.9, respectively, the difference in the lower limit of reliability obtained by different sample sizes according to three prior distributions.

| Average value all non-zero failure | Aggressive ratio all non-zero failure | $n = 5$ | | $n = 10$ | | $n = 20$ | | $n = 40$ | |
|---|---|---|---|---|---|---|---|---|---|
| prior distribution | 1 | 0.38591 | 0 | 0.51384 | 0.1260 | 0.57178 | 0.1087 | 0.60608 | 0.1034 |
| | | 0.43764 | 0.1604 | 0.51079 | 0.1526 | 0.56405 | 0.1094 | 0.60191 | 0.1034 |
| prior distribution | 2 | 0.37476 | 0 | 0.49966 | 0.1260 | 0.56375 | 0.1087 | 0.60191 | 0.1034 |
| | | 0.41983 | 0 | 0.49878 | 0.0298 | 0.55700 | 0.1094 | 0.59798 | 0.1034 |
| prior distribution | 3 | 0.37106 | 0 | 0.48928 | 0 | 0.55672 | 0.1087 | 0.59798 | 0.1034 |

Proof: $\psi(x) = \dfrac{M \cdot B(1/2,1/2)\int_x^1 R^{s-1}(1-R)^{f-1}dR + \int_x^1 R^{1/2+s-1}(1-R)^{1/2+f-1}dR}{M \cdot B(1/2,1/2)\int_0^1 R^{s-1}(1-R)^{f-1}dR + \int_0^1 R^{1/2+s-1}(1-R)^{1/2+f-1}dR}$

Where, $M > 0$, that is $Beta(0,0) = M \cdot R^{-1}(1-R)^{-1}$.

Obviously, for a given $s$ and $f$, $\psi(x)$ is a monotone decreasing function.

**Table 4   People's prior knowledge of product reliability.**

| Levels | Prior distributions |
| --- | --- |
| level 1 (high) | $Beta(0,0)$ |
| level 2 ( average) | $[Beta(0,0)+Beta(1/2,1/2)]/2$ |
| level 3 (medium) | $Beta(1/2,1/2)$ |
| level 4 (low) | $[Beta(1/2,1/2)+Beta(1,1)]/2$ |
| level 5 (unknown) | $Beta(1,1)$ |

After the above formula is simplified, there are

$$\frac{M \cdot B(1/2,1/2)\int_{R_L^1}^1 R^{s-1}(1-R)^{f-1}dR + \int_{R_L^2}^1 R^{1/2+s-1}(1-R)^{1/2+f-1}dR}{M \cdot B(1/2,1/2)\int_0^1 R^{s-1}(1-R)^{f-1}dR + \int_0^1 R^{1/2+s-1}(1-R)^{1/2+f-1}dR} = 1-\alpha$$

Then according to $R_L^1 > R_L^2$ in Eq. (1), then $\psi(R_L^1) < 1-\alpha$, $\psi(R_L^2) > 1-\alpha$.
According to $\psi(R_L^4) = 1-\alpha$, then $R_L^1 > R_L^4 > R_L^2$.
Similarly, $R_L^2 > R_L^5 > R_L^3$.

In this way, expert's prior knowledge of product reliability can be divided into the above five levels: high, average, medium, low, unknown. The corresponding prior distributions are as shown in in Table 4.

Based on the prior knowledge of experts and engineering technicians, the prior distribution can be determined more accurate by using the comprehensive evaluation method (*Balakrishnan, Hon & Navarro, 2011*; *De Luca, 2021*; *Wang et al, 2019*). There are many methods for fuzzy comprehensive evaluation. Here, the Delphi method with confidence is used as follows:

Step 1: a threshold $\lambda(1/2 < \lambda < 1)$ should be specified as the lower bound of expert opinion concentration.

Step 2: $n$ experts are invited to independently judge the product reliability according to their own experience and understanding of the product: high, average, medium, low, unknown.

Step 3: count the number of each level expert ($N_i$), and calculate $Z_i = N_i/n(i = 1,2,3,4,5)$. If there is $Z_i < \lambda$ for each expert $i$, repeat the first step and return the statistical result $(Z_1, Z_2, Z_3, Z_4, Z_5)$ to the experts for reference. Until $k$ appears, so that $Z_k \geq \lambda$.

Step 4: the expert is asked to do a final judgment, give the trust degree of the judgment $\gamma_j(j = 1,2,\ldots,n)$. Set the lower limit of the trust $\gamma_0$, and omit the opinion of the expert whose trust degree is less than $\gamma_0$. Recalculate $(Z_1, Z_2, Z_3, Z_4, Z_5)$ and record $Z = \max\{Z_1, Z_2, Z_3, Z_4, Z_5\}$, then $K$ corresponding to $Z$ is the final evaluation result. The number of experts who choose level $K$ is $N_K$, and the trust degree given by them is $\gamma_1^{(K)}, \gamma_2^{(K)}, \ldots, \gamma_{N_K}^{(K)}$, then the trust degree of the evaluation result can be expressed as:

$$\gamma = \frac{1}{N_K}\sum_{j=1}^{N_K}\gamma_j^{(K)} \tag{3}$$

## Priori information fusion method

For the traditional reliability theory (normal or failure), the prior distribution in the engineering application usually uses its conjugate prior $Beta(a,b)$, the prior density is:

$$\pi_1(R;a,b) = Beta(a,b) = \frac{1}{\beta(a,b)} R^{a-1}(1-R)^{b-1} \qquad 0 \leq R \leq 1 \qquad (4)$$

Where $a,b$ is super parameters respectively, so there are two corresponding second prior distributions, which are the uniform distributions on $[a^L, a^U]$ and $[b^L, b^U]$ respectively, and the density is:

$$\pi_{21}(a) = \begin{cases} 1/(a^U - a^L), & a^L < a < a^U \\ 0, & other \end{cases} \qquad (5)$$

$$\pi_{22}(a) = \begin{cases} 1/(b^U - b^L), & b^L < b < b^U \\ 0, & other \end{cases} \qquad (6)$$

According to the multi-layer Bayes method of binomial distribution, the multi-layer prior density of $R$ is:

$$\pi(R) = \int_{b^L}^{b^U} \int_{a^L}^{a^U} \pi_1(R;a,b).\pi_{21}(a)\pi_{22}(b)dadb = \frac{\int_{b^L}^{b^U} \int_{a^L}^{a^U} \frac{1}{\beta(a,b)} R^{a-1}(1-R)^{b-1}dadb}{(a^U - a^L)(b^U - b^L)}. \qquad (7)$$

Considering that estimation and inspection are all based on posterior distribution, when a prior distribution is available and on-site data $(n,s)$ is achieved, $n$ is the total number of tests and $s$ is the number of successful tests, the posterior density can be achieved as follows:

$$\pi(R|n,s) = \frac{R^s(1-R)^{n-s}\pi(R)}{\int_0^1 R^s(1-R)^{n-s}\pi(R)dR} = \frac{\int_{b^L}^{b^U} \int_{a^L}^{a^U} \frac{1}{\beta(a,b)} R^{a+s-1}(1-R)^{b+n-s-1}dadb}{\int_{b^L}^{b^U} \int_{a^L}^{a^U} \frac{\beta(a+s,b+n-s-1)}{\beta(a,b)} dadb} \qquad (8)$$

When the test data is failure, $a > 1$ and $0 < b < 1$ are determined according to the principle that $R$ is more likely when there is no failure, but $R$ is less likely when there is less , and the second prior is the uniform distribution on $(1,c)$ and $(0,1)$ (*Shen et al., 2023*; *Levitin, 2001*), that is

$$a^L = 1, a^U = c, b^L = 0, b^U = 1. \qquad (9)$$

If the first prior distribution is $Beta(a,b)$, the significance of the super parameter $a,b$ will be weakened, the multi-layer prior density formula Eq. (7) and the posterior density formula Eq. (8) are no longer conjugate, thus losing the advantages of the beta distribution. Therefore, the first prior distribution is the uniform distribution on $(a,1)$, and the super parameter $a$ is the lower bound of the possible value of reliability $R$.

When the on-site data and historical batch data are respectively from two different populations $X$ and $Y$, we proposed most of the historical batch data, a mixed prior is introduced. The mixed prior of the success or the failure population is:

$$\pi_\rho(R) = \rho Beta(a,b) + (1-\rho) \qquad 0 \leq \rho \leq 1 \qquad (10)$$

When evaluating the reliability of high reliability system, given the confidence degree $\gamma$, $R_L$ will be solved from Eq. (11):

$$\int_{R_L}^{1} \pi_\rho(R|s) = \gamma \tag{11}$$

## Reliability evaluation model of three-state system

For a high reliability module, assume that the module is composed of $n$ components from the functional logic, which is recorded as $N = (1, 2, \ldots, n)$; the $i$ component has multiple states, which is recorded as $M = (1, 2, \ldots, m)$; the occurrence probability of each state is recorded as $p_{ij}(i = 1, 2, \ldots, n; j = 0, 1, \ldots, m)$, and $\sum_{i=0}^{m} p_{ij}(i = 1, 2, \ldots, n)$. There are two possibilities for each state: occurrence and non-occurrence, and the probabilities are $p_{ij}$ and $1 - p_{ij}$.

For the $i$ component, the historical data is recorded as $X = (x_0, x_1, \ldots, x_m)$, and $x_j$ is the numerical value of historical data in $j$ state. On-site data is recorded as $X' = (x'_0, x'_1, \ldots, x'_m)$, and $x'_j$ is the numerical value of on-site data in $j$ state.

The on-site data and historical data are from two categories, decrease the effect of historical data and on-site data on reliability evaluation (*Zhan & Niu, 2013*; *Liu, Paulino & Gardoni, 2016*; *Bai et al., 2021*). At the same time, make the most of the information in historical data, a mixed prior is introduced:

$$\pi_\rho(p_{ij}) = \rho Beta\left(x_j, \sum_{k=0, k\neq j}^{m} x_k\right) + (1-\rho)0 \leq \rho \leq 1 \tag{12}$$

After obtaining the on-site data $Y' = \left(x'_j, \sum_{k=0, k\neq j}^{m} x'_k\right)$, the posterior density can be derived as:

$$\pi_\rho(p_{ij}|x) = \frac{MBeta\left(x'_j + 1, \sum_{k=0, k\neq j}^{m} x'_k + 1\right) + NBeta(x'_j + x_j, \sum_{k=0, k\neq j}^{m}(x'_k + x_k))}{M + N} \tag{13}$$

Where, $M = (1-\rho)\beta\left(x_j, \sum_{k=0, k\neq j}^{m} x_k\right)\beta(x'_j + 1, \sum_{k=0, k\neq j}^{m} x'_k + 1)$

$$N = \rho\beta\left(x'_j + x_j, \sum_{k=0, k\neq j}^{m}(x'_k + x_k)\right)$$

According to Eq. (11), for reliability evaluation of high reliability system, $p_{ijL}$ is solved from the following equation after the confidence $\gamma$ is given:

$$\int_{p_{ijL}}^{1} \pi_\rho(p_{ij}|x) = \gamma \tag{14}$$

# EXPERIMENTAL RESULTS AND DISCUSSION

## Reliability analysis process of three-state system

In order to effectively evaluate the reliability of the three-state system, a typical module with progressive development trend must be reasonably selected (*Xu & Chen, 2014*; *Kwon, 2014*). There are three states for a typical module: functional failure, potential failure and safety, which are recorded as $M = (0, 1, 2)$; the probabilities of occurrence and non-occurrence of each state are $p_{ij}$ and $1 - p_{ij}$ $(j = 0, 1, 2)$ respectively. For the $i$ component, the historical data is recorded as $X = (x_0, x_1, x_2)$, and $x_0, x_1, x_2$ represent the numerical value of functional failures, the numerical value of potential failures and the numerical value of safety of the historical data respectively. On-site data are recorded as $X' = (x'_0, x'_1, x'_2)$, $x'_0, x'_1$ and $x'_2$ represent the numerical value of functional failures, the numerical value of potential failures and the numerical value of safety of on-site data, respectively. If the occurrence and non-occurrence of the failure of the $i$ component obeys the binomial distribution, the historical data $X = (x_0, x_1, x_2)$ is marked as $Y = (x_0, x_1 + x_2)$, $x_0$ and $(x_1 + x_2)$ represent the fault numerical value and the fault-free numerical value in the historical data respectively. On-site data $X' = (x'_0, x'_1, x'_2)$ can be recorded as $Y' = (x'_0, x'_1 + x'_2)$, $x'_0$ and $(x'_1 + x'_2)$ represent the fault numerical value and the fault-free numerical value in the on-site data respectively.

According to formula Eqs. (12)–(14):

$$\pi_\rho(p_{i0}) = \rho Beta(x_0, x_1 + x_2) + (1 - \rho)0 \leq \rho \leq 1 \tag{15}$$

$$\pi_\rho(p_{i0}|x) = \frac{MBeta(x'_0 + 1, x'_1 + x'_2 + 1) + NBeta(x'_0 + x_0, x'_1 + x'_2 + x_1 + x_2)}{M + N} \tag{16}$$

Where, $M = (1 - \rho)\beta(x_0, x_1 + x_2)\beta(x'_0 + 1, x'_1 + x'_2 + 1)$

$N = \rho\beta(x'_0 + x_0, x'_1 + x'_2 + x_1 + x_2)$.

Therefore, the reliability calculation formula is as follows:

$$\int_{p_{i0L}}^{1} \pi_\rho(p_{i0}|x) = \gamma \tag{17}$$

## State probability of potential failure unit

In the high reliability system, it is efficient to determine the reliability of the unit structure by using $n_i - 1$ test items. Assuming that the unit reliability index without considering $j$ test items is $R_{Li,j}(j = 1, 2, \ldots, n_i)$, then rearrange $R_{Li,j}$:

$$R^*_{Li,1} \leq R^*_{Li,2} \leq \ldots R^*_{Li,j} \leq R^*_{Li,n_i} \tag{18}$$

And

$$R_{Li} \leq R^*_{Li,1} \leq R^*_{Li,2} \leq \ldots R^*_{Li,j} \leq R^*_{Li,n_i}. \tag{19}$$

From the formula above, the reliability index is $R_{Li,n_i}$, the corresponding test items have the biggest impact to the evaluation of unit reliability. Hence, the test items which correspond to $R_{Li,n_i}$ are the key tracking test links.

Huang et al. (2023), *PeerJ Comput. Sci.*, DOI 10.7717/peerj-cs.1439

For high reliability system, if the unit has the 'Safety-Potential failure-Functional failure' work mode, then the following steps can be considered:

The state probability of unit safety is the unit reliability by all test items.

The state probability of unit functional failure is the lose efficiency which not considering the tracking items.

The difference numerical value between the reliability without considering the tracking items and the reliability with considering all test items is the state probability of unit potential failure.

Then

$$
\begin{cases}
P(x_i = 1) = R_{Li} \\
P(x_i = 0.5) = R^*_{Li, n_i} - R_{Li} \\
P(x_i = 0) = 1 - R^*_{Li, n_i}
\end{cases}
\tag{20}
$$

## Result comparison and analysis

The reliability of the equipment is evaluated according to the conjugate prior and mixed prior methods, respectively. This paper adopts data from open literature (*Huang, Duan & Hao, 2010*; *Wang, Cai & Jiao, 2007*). In the process of multi-source data fusion, firstly, the detection items (performance indicators) are considered as a series system, and their reliability is calculated separately; secondly, it comprehensively evaluates the three state reliability of each module based on the Bayesian reliability evaluation method.

Method 1: according to Bayes hypothesis, the distribution interval of $R$ is $(0, 1)$.

Method 2: the distribution interval of $R$ is $(a, 1)$.

Method 3: the distribution interval of $R$ is $(a, 1)$, the expert gives the estimation interval (interval number) of $a$, and gives the confidence level of this interval number. The results of expert opinions are shown in Table 5.

A typical module is composed of power supply (D), control circuit (K), photoelectric converter (G), shaping amplifier (X), rotor (Z), amplifier (F), trigger circuit (C), counter (J) and other modules.

The photoelectric converter and the shaping amplifier are mutually compensated and backed up; the rotor, amplifier and trigger circuit form a compensation backup system. The control circuit, photoelectric converter and shaping amplifier in the system have three states respectively: 1 (safe), 0.5 (potential failure) and 0 (functional failure); The other units are in two states: 1 (safe) and 0 (functional failure). The multi-state electronic system structure with potential failure units effectively reflects the actual operation of the system, the system reliability block diagram is shown in Fig. 1.

The field test conditions of the typical module are: 8 tests, 6 failures. That is $n = 8, s = 2$. The reliability evaluation is carried out according to the above different methods. When we set the confidence to 0.9, the comparison of reliability evaluation results are shown in Table 6.

The first group of experts has a detailed understanding of the development process of the virtual prototype. They know that most of the components and subsystems of the virtual prototype have been fully tested and should have high reliability. However, because

**Table 5   Summary of expert opinions.**

| Expert No. | | 1 | 2 | 3 | 4 | 5 | 6 | 7 | 8 |
|---|---|---|---|---|---|---|---|---|---|
| Estimation interval of $a$ | lower | 0.55 | 0.65 | 0.55 | 0.75 | 0.35 | 0.45 | 0.35 | 0.45 |
| | upper | 0.85 | 0.95 | 0.95 | 0.95 | 0.75 | 0.65 | 0.75 | 0.65 |
| Trust | lower | 0.65 | 0.65 | 0.75 | 0.65 | 0.85 | 0.75 | 0.85 | 0.75 |
| | upper | 0.75 | 0.75 | 0.85 | 0.75 | 0.95 | 0.85 | 0.95 | 0.85 |

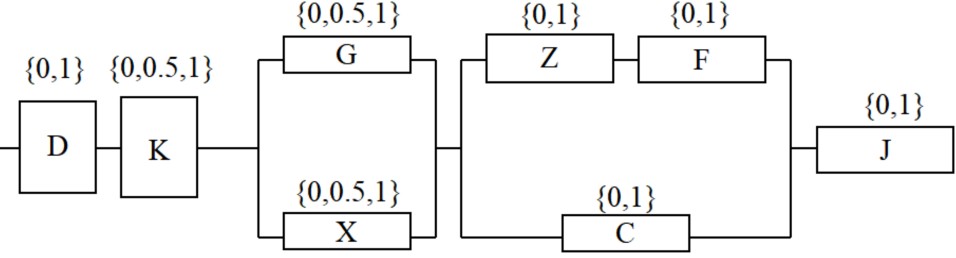

**Figure 1   Reliability block diagram of a typical module.**

**Table 6   Comparison of reliability evaluation results.**

| Method | | $a$ | $R_L$ |
|---|---|---|---|
| Method 1 | | 0 | 0.50992 |
| Method 2 | set 1 | 0.7625 | 0.77571 |
| | set 2 | 0.4875 | 0.55434 |
| | all data | 0.6250 | 0.65962 |
| Method 3 | set 1 | (0.5666, 0.8778) | 0.65622 |
| | set 2 | (0.3438, 0.6562) | 0.58619 |
| | all data | (0.4426, 0.7627) | 0.63097 |

of virtual prototype, they are not very confident about the design scheme of the virtual prototype and the assembly process of the system, the first group of experts gave a high estimate of a, but only 60% to 70% confidence. The second group of experts, based on the experience gained from the development of similar or similar products, believe that the reliability of the virtual prototype was generally 30 to 70 percent of that of the formal product, and have a high degree of confidence in this judgment.

Since it is difficult to distinguish the importance of the two groups of experts, equal weights are used in method 2. As can be seen in Table 4, the opinions of the two groups' experts are quite different. Therefore, it is difficult for the experts to agree with the reliability evaluation results achieved by averaging the experts' experience directly. In method 3, due to the use of interval numbers, the trust of experts is used as the weight to reduce the difference among the opinions of the two groups of experts. At the same time, due to the robustness of the method, the evaluation results are easily accepted by all aspects.

On the basis of the above reliability evaluation results, we adopted the method 3 to estimate the state probability of the eight constituent units of the typical module, fully

| state | D | K | G | X | Z | F | C | J |
|---|---|---|---|---|---|---|---|---|
| 1 | 1 | 0.1256 | 0.7030 | 0.7284 | 0.9930 | 0.9901 | 0.9941 | 1 |
| 0.5 | 0 | 0.8744 | 0.2909 | 0.2684 | 0 | 0 | 0 | 0 |
| 0 | 0 | 0 | 0.0060 | 0.0032 | 0.0070 | 0.0099 | 0.0059 | 0 |

**Table 7 Conditional probability of each unit state of the system.**

integrating the historical test data and field test data of each unit, and the conditional probability results of each unit are shown in Table 7. The data in the Table 7 is consistent with the actual work of this typical module.

It can be seen from the above table that when considering the potential failure state, the conditional probability of the control circuit K in the potential failure is the highest, so it is necessary to strengthen the detection of the control circuit K.

# CONCLUSIONS

Detection, classification, identification of recessive or intermittent failure features and quantitative evaluation of uncertain data are urgent problems in the field of condition monitoring of high reliability system. Aiming at the limitations of traditional Bayesian networks in dealing with the reliability problems of three-state related systems, considering the degradation process and the propagation trend of high reliability system.

Accurate analysis of the statistical characteristics of non-failure data forms a dominant modal data set, which is based on the statistical characteristics of non-failure data sources. That helps to identify the system reliability status in real time and objectively reflects the actual operation process of equipment. It is beneficial to the reliability state assessment and quantitative analysis of early faults of high reliability system. This article makes full use of the potential failure characteristics of non-failure information, and establishes a three-state system reliability evaluation model which reflects the actual work of high reliability system. The main conclusions obtained are as follows:

(1) A multi-source information fusion model was established under the condition of no prior failure information. The reasonable fusion of the historical data in the three-state system and the on-site data can effectively obtain the prior information and it can be used to evaluate lower limits of equipment reliability, which can help to evaluate the reliability state of high reliability system and predict the early failure.

(2) A reliability evaluation model for a three state system was established based on the fusion of prior data and on-site data. This method can train mathematical models by using a large amount of historical detection data, and using the tolerance between on-site data and normal data as input vectors, greatly improved the accuracy of reliability evaluation.

(3) The effectiveness of theoretical methods and mathematical models was verified through experiments. The research methods and models in this paper can be extended to the reliability sampling test of the whole life cycle management of the same type system at various stages, such as research and development, production, storage and use, so as to correctly evaluate the quality status, performance propagation trend of the system, and

obtain better economic and social benefits at a small testing expenses. The follow-up work of the paper will focus on the potential failure prediction of the three-state system.

## ACKNOWLEDGEMENTS

The authors would like to express their gratitude to the Zhuhai College of Science and Technology for their assistance throughout the research process.

### Funding

This work was supported by the Key Special Foundation of Universities in Guangdong Province, China (2020zdzx2032). The funders had no role in study design, data collection and analysis, decision to publish, or preparation of the manuscript.

### Grant Disclosures

The following grant information was disclosed by the authors:
The Key Special Foundation of Universities in Guangdong Province, China: 2020zdzx2032.

### Competing Interests

The authors declare there are no competing interests.

### Author Contributions

- Jingde Huang conceived and designed the experiments, performed the experiments, performed the computation work, authored or reviewed drafts of the article, and approved the final draft.
- Zhangyu Huang conceived and designed the experiments, performed the experiments, performed the computation work, prepared figures and/or tables, and approved the final draft.
- Xin Zhan analyzed the data, performed the computation work, prepared figures and/or tables, and approved the final draft.

### Data Availability

 The data and codes are available in the Supplementary File.

### Supplemental Information

Supplemental information for this article can be found online at http://dx.doi.org/10.7717/peerj-cs.1439#supplemental-information.

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
