# Peer review of "Research on three-state reliability evaluation method of high reliability system based on multi-source prior information"

_PeerJ Computer Science, doi:10.7717/peerj-cs.1439_

## Round 0.1 · original submission · Minor Revisions

Based on reviewers' comments the manuscript needs some revisions.

·

Basic reporting

1- Some minor grammatical errors are observed in the text (for example, the words its-reduce-system-state-supplement, system in lines 22, 24, 30, 31, 33, and 41 respectively in the abstract section) which need to be corrected.
2- The authors have drawn a conclusion from the first paragraph of the introduction (lines 78 and 79), while they used old sources before; It is necessary to examine the more recent origin. There is a similar case in lines 160 and 161.
3- It is suggested to add the research background section based on the research conducted in recent years.

Experimental design

The research question is well defined, the standards of a scientific report have been followed, complete and accurate details have been provided, and the gap identified has been filled.

Validity of the findings

The manuscript has an impact and novelty, the data is presented in an adequate form and the results are significant.

Additional comments

1- It is better for the authors to clearly state the contribution of the research in the abstract.
2- Some claims made in the text are made without the support of literature. It is necessary for the authors to use reliable and up-to-date sources (see lines 88 to 110, 124 to 129, and 148 to 155).
3- It is better to provide an authentic source for formulas taken from other studies.

·

Basic reporting

The research topic is very interesting. But it is better to correct the following items:
1- The abstract is incomplete. The main problem of the research is not clear. And you should bring at least two key results.
2- Because the Bayes hypothesis is the important issue of this research, it is better to describe the Bayes problem.
3- In the PROPOSED METHODOLOGY section, it is better. First, specify the implementation steps of the problem. It is confusing.
4- Beta value is considered for three modes. (0-0.5-1). In what conditions are they used? What does zero mean? What does the value of 0.5 mean? Which one is more suitable?
5- Many constraints and variables are not defined. Define them first.
6- The conclusion should be more complete and clear.that is weak

Experimental design

No comment

Validity of the findings

Describe the validation of the method. The given example is not enough.

Additional comments

The research topic is very interesting. But it is better to correct the following items:
1- The abstract is incomplete. The main problem of the research is not clear. And you should bring at least two key results.
2- Because the Bayes hypothesis is the important issue of this research, it is better to describe the Bayes problem.
3- In the PROPOSED METHODOLOGY section, it is better. First, specify the implementation steps of the problem. It is confusing.
4- Beta value is considered for three modes. (0-0.5-1). In what conditions are they used? What does zero mean? What does the value of 0.5 mean? Which one is more suitable?
5- Many constraints and variables are not defined. Define them first.
6- The conclusion should be more complete and clear.that is weak

---

## Round 0.2 · accepted · Accept

All the reviewers' comments have been addressed carefully and sufficiently, the revisions are rational from my point of view, I think the current version of the paper can be accepted.

·

Basic reporting

It is persuasive.

Experimental design

The standards have been met.

Validity of the findings

They are valid.

Additional comments

Kudos to the authors for executing the revisions well.

·

Basic reporting

Items are acceptable

Experimental design

Items are acceptable

Validity of the findings

Items are acceptable

Additional comments

There is no special comment